# Oxidative cyclo-rearrangement of helicenes into chiral nanographenes

Chengshuo Shen [1], Guoli Zhang[1], Yongle Ding[1], Na Yang[1], Fuwei Gan[1], Jeanne Crassous [2] & Huibin Qiu [1✉]

Nanographenes are emerging as a distinctive class of functional materials for electronic and optical devices. It is of remarkable significance to enrich the precise synthetic chemistry for these molecules. Herein, we develop a facile strategy to recompose helicenes into chiral nanographenes through a unique oxidative cyclo-rearrangement reaction. Helicenes with 7~9 *ortho*-fused aromatic rings are firstly oxidized and cyclized, and subsequently rearranged into nanographenes with an unsymmetrical helicoid shape through sequential 1,2-migrations. Such skeletal reconstruction is virtually driven by the gradual release of the strain of the highly distorted helicene skeleton. Importantly, the chirality of the helicene precursor can be integrally inherited by the resulting nanographene. Thus, a series of chiral nanographenes are prepared from a variety of carbohelicenes and heterohelicenes. Moreover, such cyclo-rearrangement reaction can be sequentially or simultaneously associated with conventional oxidative cyclization reactions to ulteriorly enrich the geometry diversity of nanographenes, aiming at innovative properties.

[1] School of Chemistry and Chemical Engineering, Frontiers Science Center for Transformative Molecules, State Key Laboratory of Metal Matrix Composites, Shanghai Jiao Tong University, Shanghai, China. [2] Univ Rennes, Institut des Sciences Chimiques de Rennes, UMR CNRS 6226, Rennes, France. ✉email: hbqiu@sjtu.edu.cn

Following the impressive development of graphene materials[1,2], nanographenes with size limited in the nanometer scale are currently emerging as prominent semiconducting substances due to their open bandgap[3], which is highly useful for electronic devices[4], energy storage[5], and sensors[6]. Conventionally, nanographenes are prepared by fragmentation of relatively large-size carbon derivatives[7,8]. Although the fabrication procedures are normally simple and ready to scale up, it is of enormous difficulty to make atomistically precise products. To this end, "bottom-up" synthetic strategies are being developed to construct nanographenes with accurate and well-tunable molecular structures[3]. Amongst, intramolecular cyclization of tailor-made dendritic oligophenylenes plays an indispensable role as a result of its high conversion and broad scope (Fig. 1a, left). This generally involves graphitization through Scholl-type oxidative cyclization[9], photocyclization[10], and HF zipping reactions[11]. On the other hand, annulative π-extension (APEX)[12,13], alkyne benzannulation[14,15], and aryne cyclotrimerization[16] reactions, either by enlarging a polycyclic aromatic core or fitting together aromatic fragments, are typical alternative pathways to complex nanographenes (Fig. 1a, right). These advances have provided an access to a vast number of nanographenes with relatively regular planar, curved, and helical geometries. However, the synthesis of more sophisticated and distinctive nanographenes remains a remarkable challenge as a consequence of the fact that most current reactions only allow the fusion of aromatic fragments into a larger system within a constant geometry regime.

Helicenes are a type of chiral polycyclic aromatic molecules constituted by *ortho*-fused benzenoid cycles[17–19]. The distorted sp²-carbon skeleton renders helicenes with relatively higher chemical activities. Thus, helicenes undergo intramolecular cyclization through Diels-Alder reaction into 3D bridge-ring frameworks[20] or deform into nearly planar structures through oxidative cyclization[21]. Recent studies by scanning tunnel microscopy showed that the skeleton of helicene could transform on metal surfaces[22,23]. This provides a unique portal to manipulate the aromatic skeletons and to construct nanographenes.

Herein, we aim to precisely harness the skeletal reconstruction of helicenes under a high-energy state generated by controlled oxidation in solution (Fig. 1b). We report the formation of chiral nanographenes through enantio-persisting oxidative cyclo-rearrangement of a variety of helicenes and study the skeleton transformation of the helicene precursors and the properties of the resulting chiral nanographenes with the assistance of theoretical calculations. This is followed by the extension of the aromatic structures through sequential or simultaneous combination with conventional oxidative cyclization reactions.

## Results and discussion

**Oxidative cyclo-rearrangement of carbohelicenes.** It has been reported previously that [5]helicene and [6]helicene derivatives undergo Scholl cyclization upon oxidation and form planar or negatively curved structures (Fig. 2a)[24,25]. Initially, we explored the oxidative reactions of primitive carbo[6]helicene **6H** in the presence of 2,3-dichloro-5,6-dicyano-1,4-benzoquinone (DDQ) and various acids (e.g., CF₃COOH, CH₃SO₃H, CF₃SO₃H) (Fig. 2b and Supplementary Table 1). However, all the attempts ended with unreacted **6H** or a mess of unidentifiable products, indicating an uncontrollable oxidation process. Then, we shifted our focus to carbo[8]helicene **8H**, a longer helicene with the two ends fully overlapping with each other. Although the reactions conducted under less acidic conditions (in the presence of CF₃COOH or CH₃SO₃H) led to similar undesired products, the mixture containing triflic acid (CF₃SO₃H) predominantly gave rise to a highly fluorescent compound (**O8H**, oxidized product for **8H**) with a yield of 49% (Fig. 2b and Supplementary Table 1). Mass spectrometry (Supplementary Fig. 1) indicated that **O8H** possessed a $C_{34}H_{18}$ chemical formula with a loss of two hydrogen atoms from the original precursor **8H** ($C_{34}H_{20}$) and ¹H NMR displayed a non-symmetric pattern other than the $C_2$-symmetric **8H** (**O8H** revealed 18 ¹H signals, while **8H** showed 10 ¹H signals due to the symmetry, Supplementary Figs. 2 and 28). X-ray diffraction of a single crystal further confirmed the formation of an unsymmetric chiral π-conjugated system in which a benzo[*ghi*]perylene moiety was fused with a [6]helicene skeleton (Fig. 2b). Further studies on the oxidation of enantiopure **8H** precursors showed that the helical chirality of **8H** was integrally transferred to **O8H**, where *P*-**8H** formed exclusively *P*-**O8H** and *M*-**8H** solely gave rise to *M*-**O8H** (Supplementary Fig. 96).

To gain more insights into this interesting oxidation reaction, we introduced a variety of substituents on the helicene skeleton and prepared 4-bromo, 4-fluoro, 4-trifluoromethyl, 4-methyl and 5,6-diphenyl substituted carbo[8] helicenes, namely, **8H_Br**, **8H_F**, **8H_CF3**, **8H_CH3**, and **8H_2py** (Fig. 2c). After treated with DDQ and CF₃SO₃H, all these precursors transformed into **O8H**-like structures. **8H_F** and **8H_Br** yielded a mixture of α structure (with the substituents finally planted on the benzo[*ghi*]perylene side) and β structure (substituents on the single-stranded helicoid side) with a ratio of 9:1 and 3:1, respectively. On the other hand, **8H_CF3** with a highly electron-withdrawing substituent formed solely β structure, while **8H_CH3** and **8H_2py** with electron-donating substituents only yielded α products.

Density functional theory (DFT) calculations on the possibly involved intermediates and transition states indicated that the above oxidation reaction may involve four major steps (Fig. 2d). First, **8H** undergoes an intramolecular oxidative cyclization (between $C_1$ and $C_{20c}$) with the assistance of DDQ and CF₃SO₃H and transforms into a cationic intermediate **8H-IM1** with a newly-formed hexagonal cycle in the center through either a radical cation or an arenium cation pathway[26,27]. Then, the cationic **8H-IM1** reforms into a spiro intermediate **8H-IM2** through 1,2-migration to release the internal strain of the highly

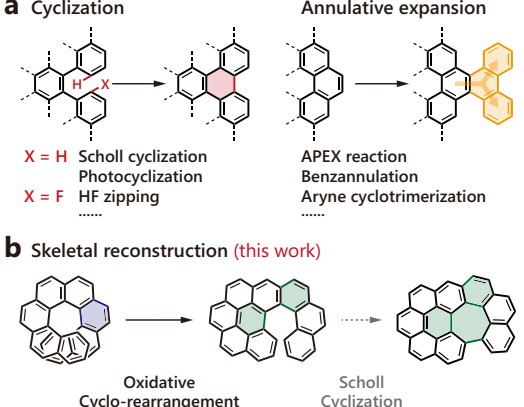

**a** Cyclization    Annulative expansion

X = H   Scholl cyclization
        Photocyclization
X = F   HF zipping

APEX reaction
Benzannulation
Aryne cyclotrimerization
······

**b** Skeletal reconstruction (this work)

Oxidative
Cyclo-rearrangement

Scholl
Cyclization

**Fig. 1 General synthetic strategies for nanographenes. a** Traditional cyclization and annulative expansion strategies. **b** Skeletal reconstruction method developed in this work.

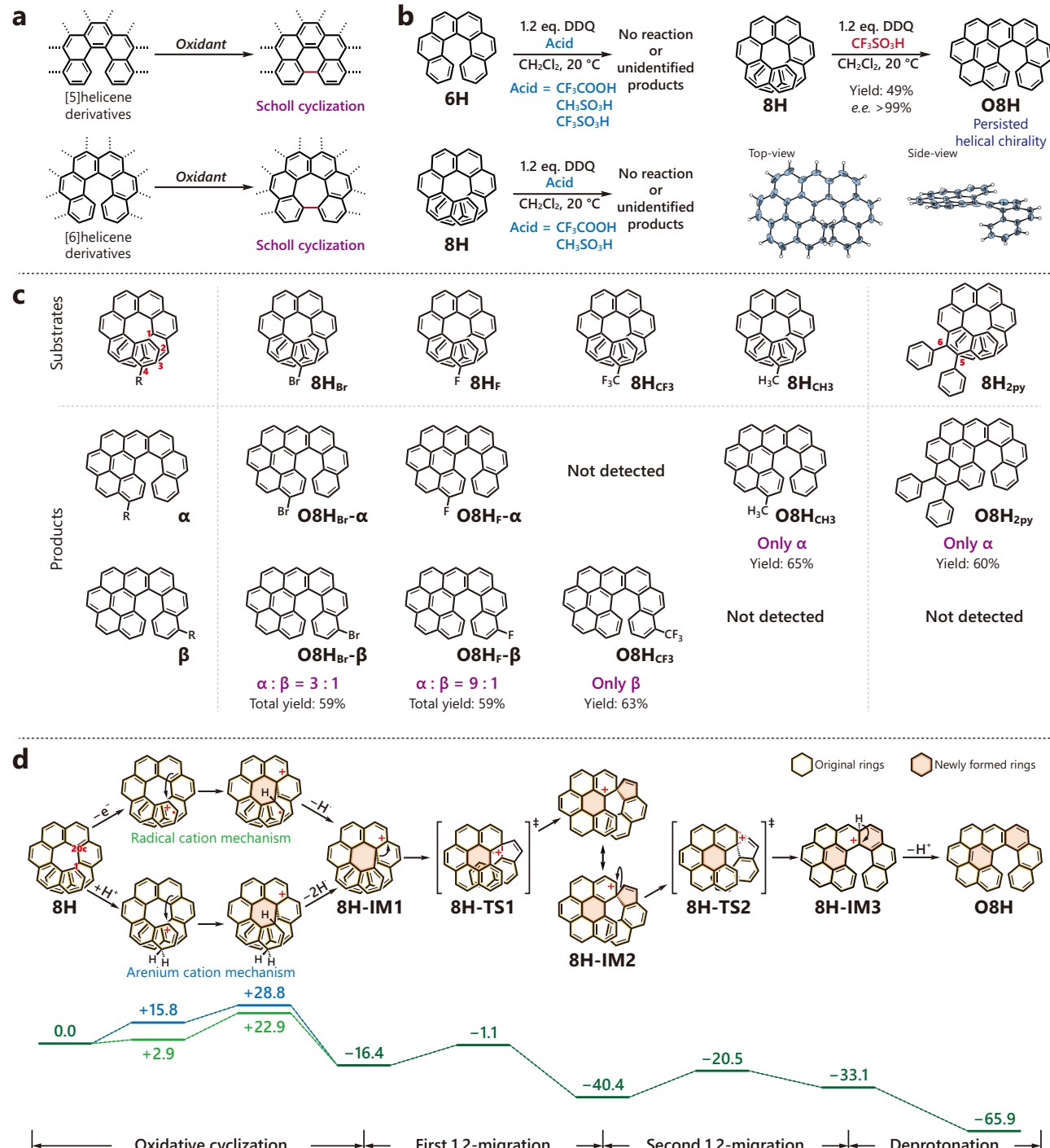

**Fig. 2 Oxidative cyclo-rearrangement of carbo[8]helicenes. a** Cyclization of [5] and [6]helicene derivatives. **b** Oxidation of primitive **6H** and **8H** by 2,3-dichloro-5,6-dicyano-1,4-benzoquinone (DDQ) in various acidic conditions. The crystal structure of **O8H** is shown in thermal ellipsoid with a 30% probability. **c** Oxidative cyclo-rearrangement of substituted carbo[8]helicenes. **d** Proposed mechanism for the oxidative cyclo-rearrangement of **8H**. Free energies are given in kcal mol$^{-1}$ and the free energy of **8H** is set as 0.

distorted structure[28]. Subsequently, **8H-IM2** transforms into a helical cationic intermediate **8H-IM3** through a second 1,2-migration via a homoallylic transition state **8H-TS2**[29,30]. Notably, the migration might undergo through a homobenzylic pathway to afford a less crowded outward product but this is probably disfavored by relatively higher activation energy (Supplementary Fig. 104). Importantly, the helical chirality of **8H** is integrally recorded by the spiro moiety in the first 1,2-migration and

intactly recovered in the second 1,2-migration, thus keeping the memory of its chirality. Finally, deprotonation of cationic **8H-IM3** leads to the formation of a relatively stable neutral product **O8H**. It appears that the electron-withdrawing substituents destabilizes the initial cation and hence favor the formation of β structures, while the electron-donating substituents stabilized the cationic moiety and thus promote the formation of α structures.

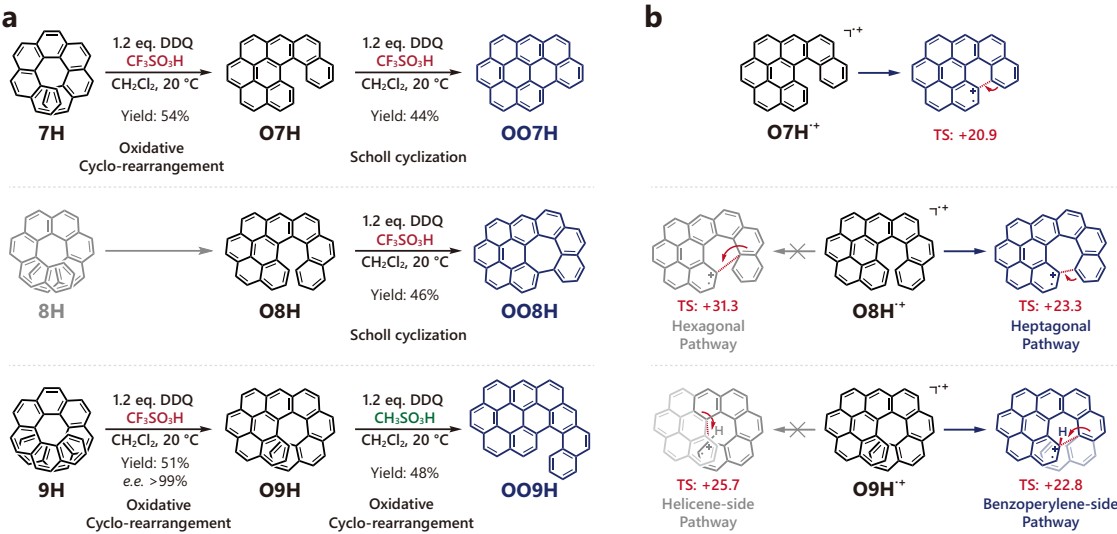

**Fig. 3 Stepwise oxidations of 7H, 8H, and 9H. a** Two-step oxidation of **7H**, **8H**, and **9H**. **b** Possible cyclization processes following a radical cation mechanism and their activation energies (free energies of the corresponding transition states, TS, in kcal mol$^{-1}$). The free energies of the radical cation intermediates **O7H**$^{\cdot+}$, **O8H**$^{\cdot+}$, **O9H**$^{\cdot+}$ are set as 0.

Further investigations showed that upon oxidation by DDQ in the presence of CF$_3$SO$_3$H, carbo[7]helicene **7H** and carbo[9] helicene **9H** also transformed into analogous structures, namely, **O7H** and **O9H** (Fig. 3a). Similarly, **O9H** efficiently inherited the absolute configuration of **9H** (Supplementary Fig. 97). Notably, upon further treatment with DDQ and acid, **O7H**, **O8H**, and **O9H** subsequently transferred into more planar structures (Fig. 3a). **O7H** readily underwent a Scholl reaction to form a fully planar structure **OO7H** (oxidized product of **O7H**, activation energy of the initial cyclization = +20.9 kcal mol$^{-1}$, Fig. 3b) like the [5]helicene derivatives. [6]helicene-like **O8H** experienced an unusual heptagonal cyclization via Scholl reaction[24,30,31] and formed negatively curved **OO8H** with twisted chirality (Fig. 5b)[32,33]. Calculations indicated that the formation of an irregular heptagon requires lower activation energy than an ordinary hexagon (+23.3 vs. +31.3 kcal mol$^{-1}$, Fig. 3b). Interestingly, [7]helicene-like **O9H** further transformed into a more π-extended structure **OO9H** containing a coronene core and a [5] helicene moiety through a second oxidative cyclo-rearrangement under a weaker acidic condition (using CH$_3$SO$_3$H), which occurred solely at the benzoperylene side probably as a result of a lower activation energy barrier (Fig. 3b). Besides, both **OO7H** and **OO8H** can be directly synthesized in one pot starting from **7H** and **8H** in the presence of more DDQ in yields of 46% and 40%, respectively.

The fusion of the benzo[*ghi*]perylene and helicene moieties in an absolute chiral fashion provides a distinctive platform to tune the properties of nanographenes. Most intuitively, these molecules showed a combined aromaticity nature of benzo[*ghi*] perylene and helicene according to the harmonic oscillator model of aromaticity (HOMA)[34] (Fig. 4a) and the nucleus-independent chemical shifts (NICS)[35] (Fig. 4b). Typically, **O7H**, **O8H**, and **O9H** showed high aromaticity in rings B, C, and F as a benzo[*ghi*] perylene fashion, and also in the terminal ring (H, I, and J, respectively) as a helicene feature. Besides, the newly formed heptagonal cycle of **OO8H** presented a typical anti-aromaticity nature (HOMA = −0.35, and NICS(0) = 12.2). Regarding the optical properties, **O7H**, **O8H**, and **O9H** displayed intense benzo [*ghi*]perylene-type absorption bands with several vibronic

progression in the region of 400–500 nm (Fig. 4c). These bands can be majorly attributed to the HOMO → LUMO transitions with the optical bandgaps of 2.59, 2.54, and 2.40 eV, respectively, which are comparable with the electrochemical bandgaps (2.54, 2.49, and 2.41 V, respectively, Fig. 4f). The enantipure **O8H** and **O9H** molecules revealed intense mirror-image electronic circular dichroism (ECD) signals with typical helicene characteristics, where for the *P* enantiomers, positive Cotton effects were formed in a longer wavelength region (350–500 nm), and negative ones were formed in shorter wavelengths (250–350 nm) (Fig. 4c). These molecules displayed blue to yellow emissions (Fig. 4e), and the enantipure **O8H** and **O9H** products showed intense circularly polarized luminescence (CPL) with prominent dissymmetric factors $g_{lum}$ values[36,37] of ±0.0055 and ±0.012, respectively (Fig. 4d).

**Oxidative cyclo-rearrangement of heterohelicenes.** Heteroatom-doping is an efficient and practical method to tailor the physical and chemical properties of polycyclic aromatic molecules[38,39]. To this end, we prepared sulfur-doped or/and nitrogen-doped methylthio[8]helicene **8H$_{mt}$**, aza[8]helicene **8H$_a$**, and thioaza[8] helicene **8H$_{ta}$**. In a typical oxidative cyclo-rearrangement condition with one equivalent of DDQ, we observed that **8H$_{mt}$** transformed not only into an oxidized product **O8H$_{mt}$** but also into a further oxidized product **OO8H$_{mt}$** (Fig. 5a). **O8H$_{mt}$** showed an **O8H**-like structure with the thiophene cycle located at the single-stranded helicoid end, while **OO8H$_{mt}$** displayed a negatively curved feature caused by the heptagonal cycle. Due to the doping of sulfur, **OO8H$_{mt}$** revealed considerably bathochromic-shifted absorption and emission bands compared to **OO8H** (Fig. 5b, c). **8H$_a$** and **8H$_{ta}$** also transformed into the rearranged products **O8H$_a$** and **O8H$_{ta}$**, respectively (Fig. 5d, e). It should be noted that protonation may occur to the nitrogen atoms, but obviously, this did not hinder the following oxidative cyclo-rearrangement, and the products can be neutralized by NaOH at the end. This protonation chemistry was further utilized to switch the chiroptical properties of enantipure **O8H$_a$** (Supplementary Fig. 98, Fig. 5f, g), and remarkably the protonated product **O8H$_a$H$^+$** showed a

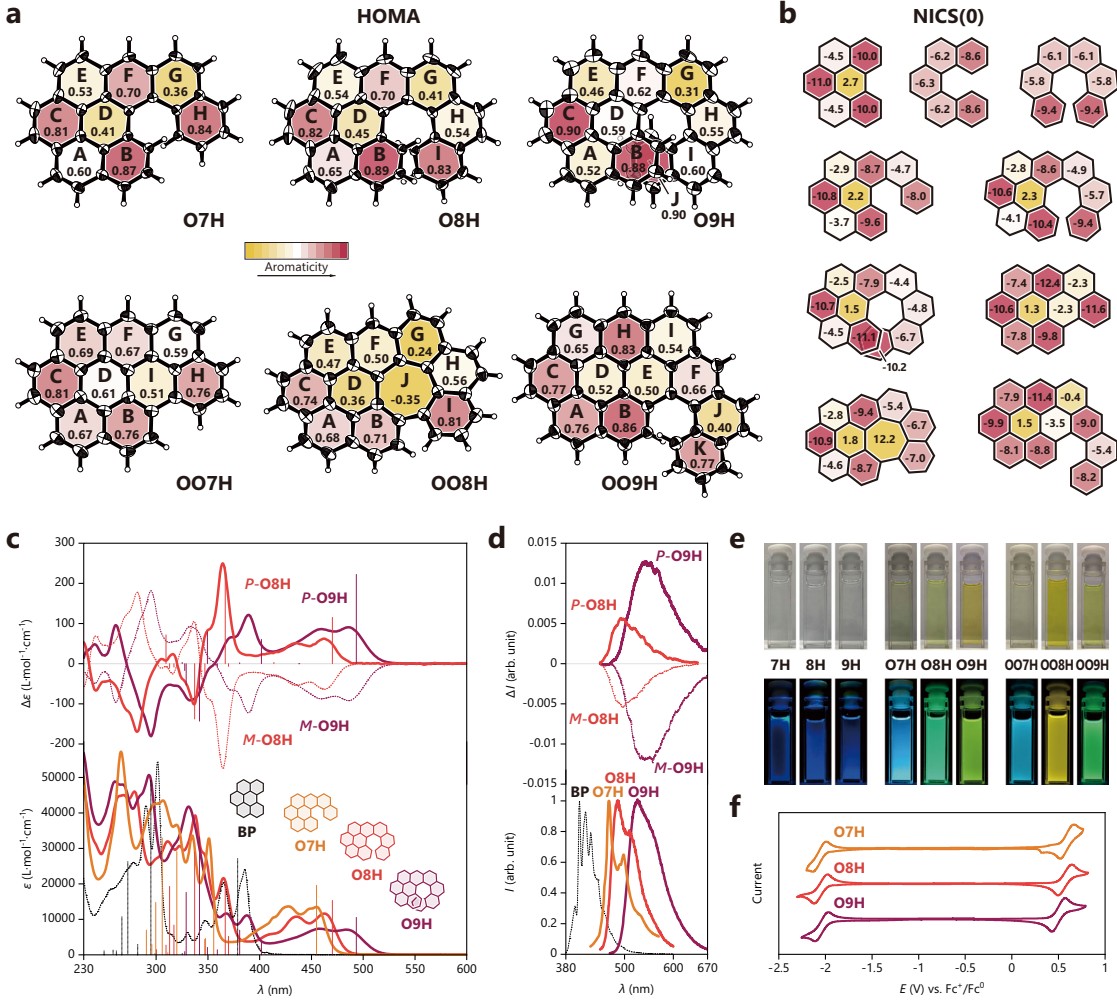

**Fig. 4 Molecular properties of O7H, O8H, O9H, OO7H, OO8H, and OO9H. a**, **b** Local aromaticity evaluation studied by harmonic oscillator model of aromaticity (HOMA) from the crystal structures (shown in thermal ellipsoid with a 30% probability) and nucleus-independent chemical shifts, NICS (0) from theoretical calculations. **c** UV-vis absorption (bottom) and electronic circular dichroism (ECD) (top) spectra of benzo[*ghi*]perylene (BP), **O7H**, **O8H**, and **O9H** (in $CH_2Cl_2$, $c = 2.0 \times 10^{-5}$ mol $L^{-1}$). Calculated excitation energies and oscillator/rotatory strengths (velocity form) are displayed as sticks (the first 10 excitations). **d** Fluorescence (bottom) and circularly polarized luminescence (CPL) (top) spectra of benzo[*ghi*]perylene, **O7H**, **O8H**, and **O9H** (in $CH_2Cl_2$, $c = 2.0 \times 10^{-5}$ mol $L^{-1}$). **e** Photographs of corresponding solutions in $CH_2Cl_2$ recorded under ambient light (top) and irradiation at 365 nm (bottom). **f** Cyclic voltammograms for **O7H**, **O8H**, and **O9H** (in $CH_2Cl_2$, containing 0.2 mol $L^{-1}$ of $n$-Bu$_4$NPF$_6$, scan rate = 0.1 mV s$^{-1}$).

significantly bathochromic-shifted CPL compared to the neutral product **O8H$_a$** (orange vs. green).

**Fabrication of more sophisticated nanographenes**. The oxidative cyclo-rearrangement reactions were ultimately associated with other reactions to fabricate more sophisticated nanographenes with distinctive geometries and properties. We first prepared an oligophenylene-like helicene derivative **6H$_{DPT}$** with a diphenyltriphenylene (DPT) group substituted on position 2 of **6H**. Upon treatment with an excessive amount of DDQ in the presence of $CF_3SO_3H$, **6H$_{DPT}$** directly transformed to **O6H$_{DPT}$** which showed a [6]helicene moiety fused to a slightly arched π-extended structure with a length of ca. 1.5 nm (Fig. 6a, b)[40]. This was probably generated by the formation of a [8]helicene moiety at first via Scholl cyclization followed by an oxidative cyclo-rearrangement at the more π-extended side. The highly extended π-system rendered **O6H$_{DPT}$** with a strong absorption band around 490 nm and an

intense greenish-yellow fluorescence around 510 nm (Fig. 6c). We also synthesized another bromo-substituted product **O8H'$_{Br}$** by treating 2-bromo[8]helicene (**8H'$_{Br}$**) with DDQ and $CF_3SO_3H$ and subsequently fused it with a perylene dii-mide (PDI) moiety through boration and Suzuki coupling. Oxidative photocyclization of the resulting product **O8H'$_{PDI}$** finally afforded a dark brown product **CO8H'$_{PDI}$** (Fig. 6d). Single-crystal analysis revealed that **CO8H'$_{PDI}$** possessed a bilayer conjugated structure[40–42] with the benzo[*ghi*]perylene and PDI moieties fused by a [8]helicene bridge (Fig. 6e). The two layers showed a typical AB stacking pattern with a short interlayer distance of 3.4 Å. The unsymmetric bilayer structure presented a strong intramolecular charge transfer complexa-tion with the HOMO located at the benzo[*ghi*]perylene layer and the LUMO at the PDI layer (Fig. 6g), leading to a broad absorption band from ca. 570 to 710 nm (Fig. 6f). Besides, the **CO8H'$_{PDI}$** molecules were found to be tightly piled up in the crystals with the PDI moiety of one molecule and the benzo[*ghi*]perylene fragment of the neighboring molecule

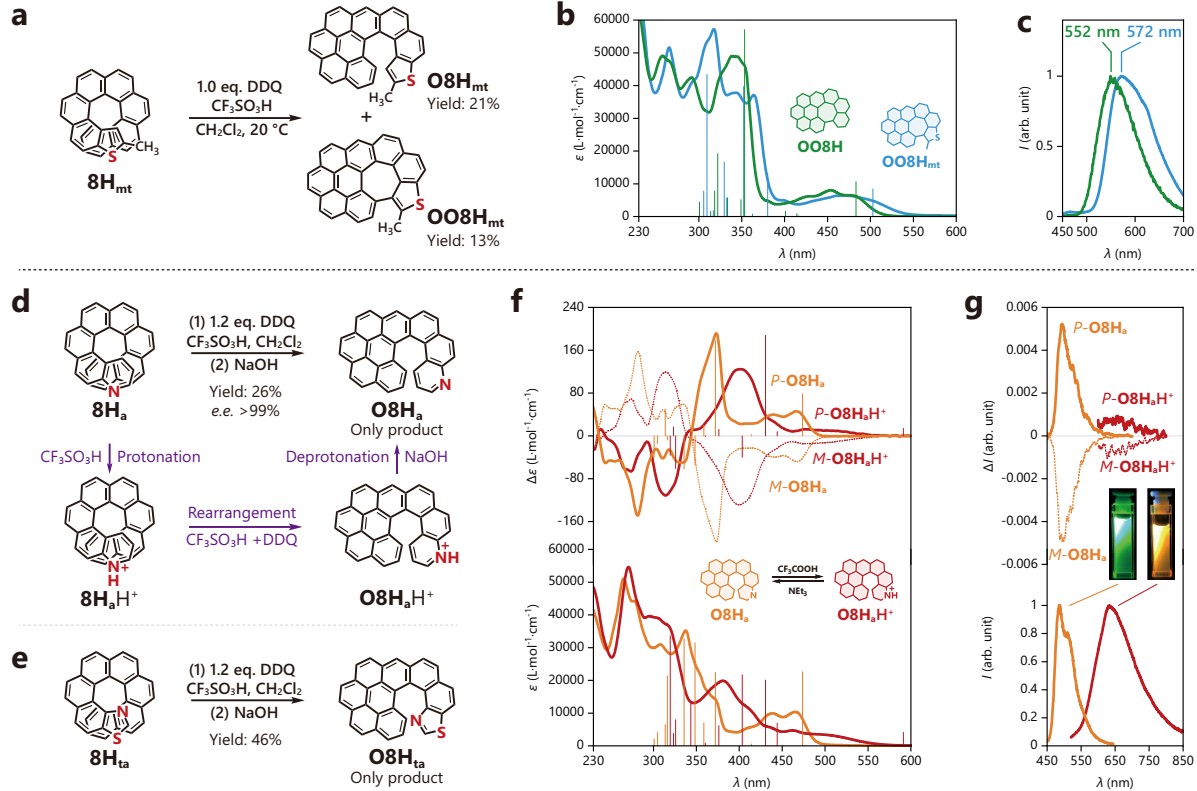

**Fig. 5 Oxidative cyclo-rearrangement of hetero-atom-doped helicenes. a** Synthesis of sulfur-doped **O8H_mt** via oxidative cyclo-rearrangement of **8H_mt**, and further cyclization to **OO8H_mt**. **b** UV-vis absorption and **c** fluorescence spectra of **OO8H_mt** with comparison to **OO8H** (in $CH_2Cl_2$, $c = 2.0 \times 10^{-5}$ mol $L^{-1}$). **d** Synthesis of nitrogen-doped **O8H_a** via oxidative cyclo-rearrangement of **8H_a**. **e** Synthesis of sulfur-doped and nitrogen-doped **O8H_ta** via oxidative cyclo-rearrangement of **8H_ta**. **f** UV-vis absorption (bottom) and electronic circular dichroism (ECD) spectra (top) of **O8H_a** and **O8H_a**H$^+$ (in $CH_2Cl_2$, $c = 2.0 \times 10^{-5}$ mol $L^{-1}$). **g** Fluorescence (bottom) and circularly polarized luminescence (CPL) spectra (top) of **O8H_a** and **O8H_a**H$^+$ (in $CH_2Cl_2$, $c = 2.0 \times 10^{-5}$ mol $L^{-1}$). The inset photographs are recorded under irradiation at 365 nm. All calculated excitation energies and oscillator/rotatory strengths (velocity form) are displayed as sticks (the first 10 excitations).

packed in an AB stacking manner with a close distance of ca. 3.6 Å (Fig. 6e).

In summary, we have developed a distinctive oxidative cyclo-rearrangement method to recompose the skeleton of helicenes to precisely fabricate chiral nanographenes with unique topologies and absolute configurations integrally inherited from their helical precursors. The facile introduction of a variety of substituents at the periphery and heteroatoms in the aromatic core, along with the sequential oxidative cyclo-rearrangement reactions, have provided diverse access to a rich array of nanographenes. The flexible association with other conventional oxidative cyclization reactions has further allowed the designable fabrication of more sophisticated chiral nanographenes. Overall, this work demonstrates a unique route to activate and reform a highly distorted aromatic system (e.g., helicenes) into flattened but lateral extended derivatives, which is virtually driven by the gradual release of the internal strains. The design of analogous energy-reservoir precursors together with controlled skeletal reconstruction reactions may open up an innovative pathway for future nanographene synthesis.

## Methods

**Typical oxidative cyclo-rearrangement reaction**. 21.4 mg of **8H** (0.05 mmol) and 13.6 mg of DDQ (0.06 mmol, 1.2 equiv.) were suspended in 20 mL of dried $CH_2Cl_2$ under argon at 20 °C, and 0.1 mL of $CF_3SO_3H$ was added to the suspension dropwise with vigorous stirring. The mixture was stirred for 30 min and then

quenched by 5 mL of saturated $NaHCO_3$ aqueous solution. The mixture was extracted by $CH_2Cl_2$ and the organic layer was washed with water twice and dried over $Na_2SO_4$. The solvent was removed at reduced pressure and the residue was purified by silica column chromatography (heptane/$CH_2Cl_2$ = 10/1, $v/v$) to afford **O8H** as orange solid (10.5 mg, yield 49%).

**Theoretical calculations**. DFT calculations were carried out using Gaussian 09 program. Geometrical optimization calculations were carried out at the PBE0-D3/def2-SVP level without any symmetry assumptions unless otherwise stated. The Cartesian coordinates of the optimized geometries are provided in Supplementary Data 1. Harmonic vibration frequency calculations were performed at the same level for obtaining the thermodynamic energies and verifying the resulting geometries as local minima (with all the frequencies real) or saddle points (with only one imaginary frequency). The assignment of the saddle points was performed using the intrinsic reaction coordinate (IRC) calculations. Nucleus-independent chemical shifts (NICS) values were calculated at the GIAO-PBE0/def2-SVP level on the optimized structures. Time-dependent density functional theory (TD-DFT) calculations were performed at the PBE0-D3/def2-SVP level on the optimized geometries for the lowest 200 vertical singlet electronic excitations. No special shift or scaling was applied unless otherwise mentioned. Isosurfaces of molecular orbitals (MOs) were drawn using the IQmol program with the isovalue set to 0.1. All calculations were performed under the SMD continuum solvent model for $CH_2Cl_2$.

## Data availability

The authors declare that all the data supporting the findings of this study are available within the paper and Supplementary Information files, and also are available from the corresponding authors upon reasonable request. The X-ray crystallographic coordinates for structures reported in this study have been deposited at the Cambridge Crystallographic Data Centre (CCDC), under deposition numbers 2049082 (**9H**), 2049080 (**8H_CF3**), 2049081 (**8H_F**), 2049079 (**7H**), 2049136 (**O7H**), 2049140 (**O8H**),

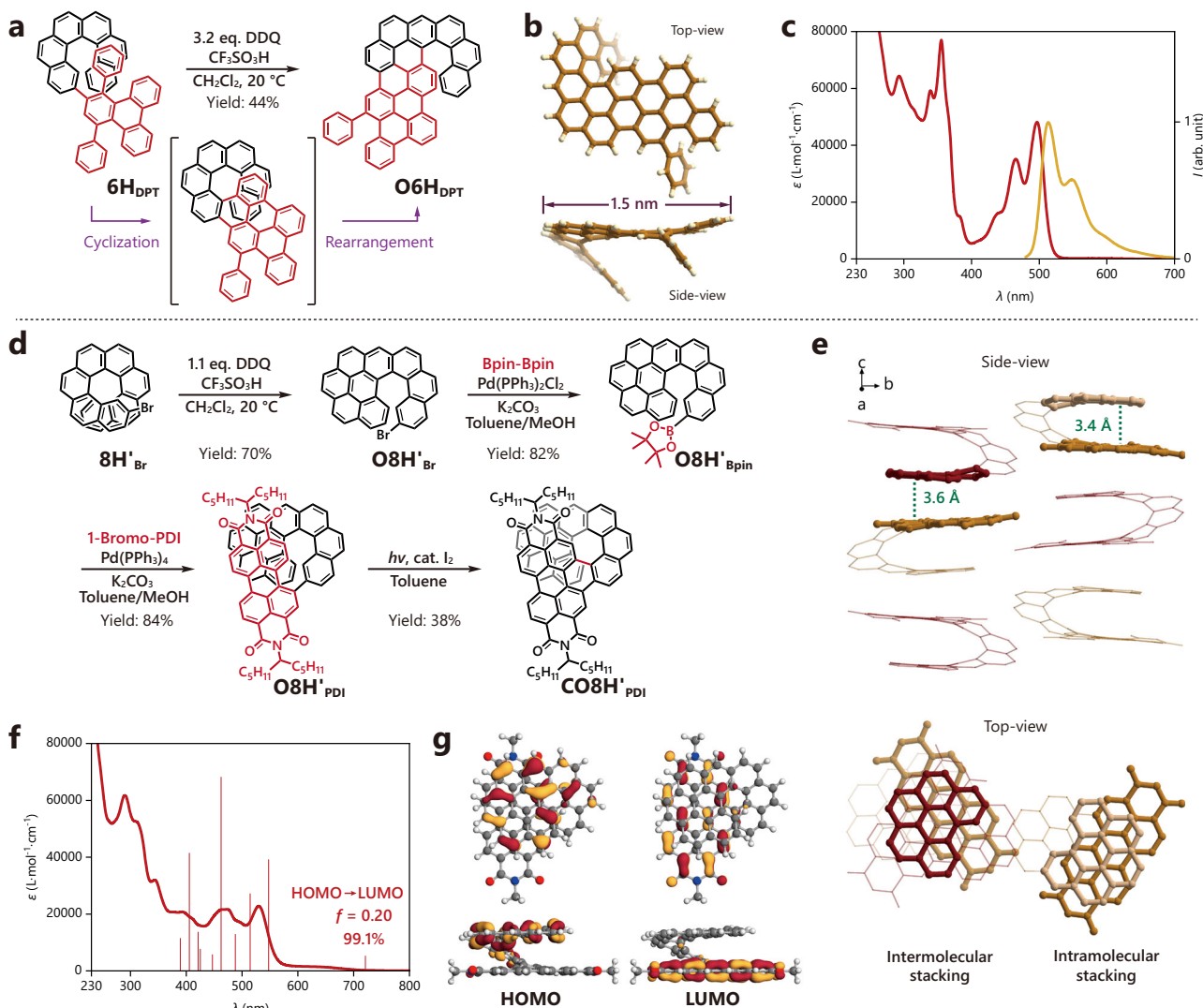

**Fig. 6 Synthesis of more sophisticated nanographenes. a** Synthesis of **O6H_DPT** through the association of oxidative cyclo-rearrangement with Scholl cyclization. **b** Crystal structure of **O6H_DPT**. **c** UV-vis absorption (red) and fluorescence (yellow) spectra of **O6H_DPT** (in $CH_2Cl_2$, $c = 2.0 \times 10^{-5}$ mol $L^{-1}$). **d** Synthesis of **O8H'_Br** via oxidative cyclo-rearrangement and sequential formation of **CO8H'_PDI** via further fusion with a PDI moiety. **e** Stacking of **CO8H'_PDI** molecules in a crystal (alkyl side chains and hydrogen atoms are omitted for clarity). **f** UV-vis absorption spectrum of **CO8H'_PDI** (in $CH_2Cl_2$, $c = 2.0 \times 10^{-5}$ mol $L^{-1}$). Calculated excitation energies and oscillator strengths of **CO8H'_PDI-CH3** (the alkyl side chains are replaced with methyl groups for simplicity) are displayed as sticks (the first 10 excitations). **g** HOMO and LUMO of **CO8H'_PDI-CH3**.

2049571 (**O9H**), 2049086 (**O8H_CH3**), 2049085 (**O8H_CF3**), 2049087 (mixed crystal of compounds **O8H_F-α** and **O8H_F-β**), 2049084 (mixed crystal of compounds **O8H_Br-α** and **O8H_Br-β**), 2049137 (**O8H_2py**), 2049572 (**OO7H**), 2049574 (**OO8H**), 2049575 (**OO9H**), 2049088 (**O8H_mt**), 2049573 (**OO8H_mt**), 2049083 (**O8H_a**), 2049089 (**O8H_ta**), 2049135 (**O6H_DPT**), 2049139 (**O8H'_Br**), 2049138 (**O8H'_Bpin**), 2049134 (**CO8H'_PDI**). These data can be obtained free of charge from The Cambridge Crystallographic Data Centre via www.ccdc.cam.ac.uk/data_request/cif.

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

## Acknowledgements

This work was financially supported by the National Key R&D Program of China (2020YFA0908100), the National Natural Science Foundation of China (21704063, 92056110, 22075180), the Innovation Program of Shanghai Municipal Education Commission (202101070002E00084), and the Science and Technology Commission of Shanghai Municipality (18JC1415500, 195271040, 20JC1415000). The authors thank Prof. Shunai Che's group in Tongji University for assistance with CPL measurement.

## Author contributions

C.S. and H.Q. conceived the project. C.S. and G.Z. designed the experiment and synthesized all the molecules. C.S., G.Z., Y.D., and F.G. performed the single-crystal X-ray diffraction experiments and data analysis. C.S. and G.Z. conducted the spectroscopic experiments and data analysis. C.S., N.Y., and G.Z. conducted the electrochemistry experiments. C.S. performed all the theoretical calculations. C.S. and H.Q. wrote the manuscript with inputs from J.C. All the authors discussed the results and commented on the manuscript.

## Competing interests

The authors declare no competing interests.
