## [Peer Review File · Nature Communications]

REVIEWER COMMENTS

Reviewer #1 (Remarks to the Author):

Shen et al. reported about synthesis chiral nanographenes from helicenes. In their work, they performed oxidative cyclization reactions to obtain chiral nanographenes and characterized their aromaticity and optical properties.

First, they demonstrated oxidative cyclization of reactions from carbo[8]helicene and its derivatives and discussed the mechanism of reaction based on the results of the energy calculation by DFT.

Subsequently, they conducted the same reactions using carbo[7]helicene and carbo[9]helicene to evaluate their proposed mechanism of reactions. Further oxidization of these helicenes was also reported. They characterized their synthesized compounds and indicated that the helical chirality of the starting material is preserved in the products. Moreover, they demonstrated the synthesis of functional nanographenes using heteroatom-doping or substitution of a bulky functional group in helicenes and characterized their spectroscopic and structural features.

The authors' synthetic strategy using helicenes for chirality control of nanocarbons is a well-considered approach with the aid of the reported work (references 21 and 22). I have evaluated that this manuscript reports two notable insights. One is the authors discuss the reaction mechanism for oxidative cyclo-rearrangement with showing real synthetic results starting from substituted or heteroatom-doped helicenes. The results of this self-assessment are increasing the effectiveness of their proposed synthetic concept. The other is the authors provide synthetic examples of chiral nanocarbons, i.e., helically-twisted graphene nanoribbons. The demonstration of preparing chiral nanocarbons using chirality of helicene will be useful for readers who will utilize the authors' synthetic strategy.

The analytical studies for characterizing products are technically sounds. In consequence, This manuscript will attract broad interest from the chemistry and materials science communities and provide benefit to the reader of Nat. Commun. Since there are some minor unclear descriptions in this manuscript, I recommend acceptance of this manuscript for publication in Nat. Commun. after addressing the following comments:

1. The authors selected carbo[6]helicene as a starting material in their first trial and failed to take control of the reaction. Taking into account such a negative result will lead to a further rationalization of the chemical reaction. I recommend the authors to discuss the reaction/un-reaction of carbo[6]helicene.
2. At line 13 on page 5, the authors describe "8Hmt transformed into an oxidized product O8Hmt and a further oxidized product OO8Hmt at the same time". Do the two products (O8Hmt and OO8Hmt) are really generated at the same time? Figure 3 shows that OO8H is synthesized with the stepwise oxidation of 8H through O8H. The given description can also be interpreted as synthesizing OO8H from 8H directly. A more clear-cut expiation about this reaction is needed.
3. Why did the helical chirality of 7H, OO8H, OO9H have not been determined by X-ray crystallography? The performed collection of the full data set using CuK α radiation seems to be appropriate for absolute structure determination. Did the authors merge Friedel pairs in data processing? I requested the authors

to explain the reason for it.

4. In CheckCIF outputs, level A or B alerts are found in O9H, O8HCF3, OO9H, OO8Hmt, and O6HDPT. The authors should address these alerts by writing vrf in CIFs.

5. Hall symbol in CIF of 8HF and OO8Hmt is notified as unsuitable in CheckCIF. Modification or explanation (using vrf) is required.

Reviewer #2 (Remarks to the Author):

This is a wonderful article that shows genuine novelty and remarkable significance to the field of curved polycyclic aromatics. The authors have found that oxidation of [n]helicenes (n = 7 to 9) with DDQ at presence of CF₃SO₃H enables unprecedented skeletal rearrangement resulting pi-expanded [m]helicenes (m = 5 to 7). The pi-expanded [6]helicenes, such as O8H, can be further oxidized by DDQ/CF₃SO₃H giving cyclized products with formation of a seven-membered ring; while pi-expanded [7]helicenes, such as O9H, can be further oxidized by DDQ/CH₃SO₃H enabling another oxidative cyclo-rearrangement. A reasonable mechanism for this oxidation cyclo-rearrangement has been proposed on the basis of DFT calculations on the possible reaction pathways. Another very interesting finding is that the chirality of the helicene precursor is transferred to the resulting pi-expanded helicene with ee higher than 99%. Moreover, this is a very solid study having all the products unambiguously characterized. It is very impressive that 23 single crystal structure have been reported in this manuscript!

The Scholl reaction is a powerful method for synthesis of polycyclic aromatics enabling formation of multiple carbon-carbon bonds in a single step. However, synthetic organic chemists still lack general ability to accurately predict whether and how the Scholl reaction will occur on a new substrate. The oxidative cyclo-rearrangement under the Scholl reaction conditions discovered here has provided important new insights to the Scholl reaction.

Therefore, this manuscript is highly recommended for publication in Nature Communication. The following minor issues should be taken into consideration for revision.

1. In Figure 1a, "Benzanullation" should be "Benzannulation".

2. OO8H is a chiral molecule and its single crystal structure contains only the M-enantiomer. Can the authors comment on the chirality transfer from O8H to OO8H in accompany with formation of a seven-membered ring?

3. Regarding the proposed mechanism shown Figure 2d, the authors are strongly suggested to explain why the second 1,2-migration occurs via a homoallylic transition rather than a homobenzylic transition. This explanation should be included in the main text rather than in the Supporting Information (Figure S104)

Reviewer #3 (Remarks to the Author):

The manuscript entitled 'Oxidative Cyclo-Rearrangement of Helicenes into Chiral Nanographenes', presents a novel skeletal reconstruction reaction in helicenes and affords straightforward access to an array of chiral polycyclic aromatic hydrocarbons.

This work could contribute to a renewed perspective of Organic Synthesis and have direct implications in overlapping fields, such as Materials Science. This kind of intramolecular rearrangements are usually poorly explored, and often remain anecdotic. Further study and understanding of these processes will provide more efficient synthetic tools. The authors provided solid evidence for their claims. Particularly, I appreciate that X-ray structure of several compounds are included.

The manuscript is clear and well written. In my opinion, it should be published in Nat. Commun. once the following minor points have been addressed or considered.

Introduction

(1) In my opinion Aryne Chemistry is among the most used methods for the synthesis of structurally complex polycyclic aromatic hydrocarbons (PAHs) and helicenes. Therefore, I would suggest the addition of an explicit mention to that approach in the introduction.

(2) In the sentence "or deform into planar structures through oxidative cyclization" I would add "nearly planar", given that reference 20 is entitled "On-surface synthesis of a nonplanar porous nanographene".

Results and discussion

(3) At the beginning of this part, authors explain how they discovered this interesting reaction, although there is not a clear optimisation. I would like to know how temperature, time, and the use of other solvents, affect to the oxidative cyclo-rearrangement (It would be great the addition of a table with few entries in the supporting information).

(4) "Mass spectroscopy" should be corrected to "Mass spectrometry".

(5) In the second paragraph of page 4. The authors say "Besides, both OO7H and OO8H can be directly synthesized in one pot starting from 7H and 8H in the presence of more DDQ." I would include the yield for each one pot transformation in the main text. These relevant results are currently available in the supporting information (46 and 40% respectively).

(6) In the last paragraph of page 5. In the sentence "The oxidative cyclo-rearrangement reactions were ultimately associated with other reactions to fabricate more sophisticated nanographenes with distinctive geometries and tailored properties." I would delete tailored.

(7) In the same paragraph, the word "nanoribbon" it is used referring to a part of compound O6HDPT. From my point of view, it would be more appropriate the use of another term, such as structure.

References

(8) Reference 31. The name of the last author should be corrected (Campaña, instead of Caompañia).

Figure 2b

(9) I would replace “Unidentical” Products with “Unidentified” Products in both cases.

Supporting Information

(10) The characterisation provided is enough to strongly support the work. However, they should report which technique was used for mass analysis in each case (MALDI, APCI, EI, etc.).

Response to Reviewers

Reviewer 1:

Shen et al. reported about synthesis chiral nanographenes from helicenes. In their work, they performed oxidative cyclization reactions to obtain chiral nanographenes and characterized their aromaticity and optical properties.

First, they demonstrated oxidative cyclization of reactions from carbo[8]helicene and its derivatives and discussed the mechanism of reaction based on the results of the energy calculation by DFT. Subsequently, they conducted the same reactions using carbo[7]helicene and carbo[9]helicene to evaluate their proposed mechanism of reactions. Further oxidization of these helicenes was also reported. They characterized their synthesized compounds and indicated that the helical chirality of the starting material is preserved in the products. Moreover, they demonstrated the synthesis of functional nanographenes using heteroatom-doping or substitution of a bulky functional group in helicenes and characterized their spectroscopic and structural features.

The authors' synthetic strategy using helicenes for chirality control of nanocarbons is a well-considered approach with the aid of the reported work (references 21 and 22). I have evaluated that this manuscript reports two notable insights. One is the authors discuss the reaction mechanism for oxidative cyclo-rearrangement with showing real synthetic results starting from substituted or heteroatom-doped helicenes. The results of this self-assessment are increasing the effectiveness of their proposed synthetic concept. The other is the authors provide synthetic examples of chiral nanocarbons, i.e., helically-twisted graphene nanoribbons. The demonstration of preparing chiral nanocarbons using chirality of helicene will be useful for readers who will utilize the authors' synthetic strategy.

The analytical studies for characterizing products are technically sounds. In consequence, This manuscript will attract broad interest from the chemistry and materials science communities and provide benefit to the reader of Nat. Commun. Since there are some minor unclear descriptions in this manuscript, I recommend acceptance of this manuscript for publication in Nat. Commun. after addressing the following comments:

We are very grateful for the reviewer's kind remarks and the following suggestions.

1. The authors selected carbo[6]helicene as a starting material in their first trial and failed to take control of the reaction. Taking into account such a negative result will lead to a further rationalization of the chemical reaction. I recommend the authors to discuss the reaction/un-reaction of carbo[6]helicene.

We thank reviewer 1 for this important comment. In our initial attempts, we have carefully screened the reaction condition for the oxidative cyclo-rearrangement of carbo[6]helicene **6H** and carbo[8]helicene **8H** and the details are listed below (Figure for review 1).

Table for review 1. Screening of reaction condition for the oxidative cyclo-rearrangement of **6H** and **8H** using TLC (heptane/CH₂Cl₂ = 5/1, v/v). Dot 1 for the raw product and dot 2 for the starting helicene.

	6H	8H
CF₃COOH	 Sunlight 365 nm 254 nm No reaction	 Sunlight 365 nm 254 nm No reaction
CH₃SO₃H	 Sunlight 365 nm 254 nm A string of unidentifiable products	 Sunlight 365 nm 254 nm A string of unidentifiable products
CF₃SO₃H	 Sunlight 365 nm 254 nm A string of unidentifiable products	 Sunlight 365 nm 254 nm Mainly O8H

Briefly, the oxidation of **6H** gave rise to a string of unidentifiable spots with multicolor luminescence when CH₃SO₃H or CF₃SO₃H was used, while **8H** yielded only one green luminescent spot (later was proven to be **O8H**) in the presence of CF₃SO₃H. We then analyzed the raw products derived from **6H** using MALDI-HR-MS (Figures for review 2 and 3). From the spectra, no obvious peaks were found around $m/z = 326$ (hexa[7]circulene), which was in agreement with the previous work where carbo[6]helicene failed to undergo dehydrogenation cyclization (Scholl reaction) to form hexa[7]circulene (*J. Org. Chem.* **1975**, *40*, 3398). Besides, the presence of high m/z species strongly indicated the onset of intermolecular coupling. However, the crude yields for these products were quite low and hence they were hard to be separated and identified for further studies. These results have been added to the revised SI.

Figure for review 2. MALDI-HR-MS spectrum of raw products derived from **6H** in the presence of $\text{CH}_3\text{SO}_3\text{H}$.

Figure for review 3. MALDI-HR-MS spectrum of raw products derived from **6H** in the presence of $\text{CF}_3\text{SO}_3\text{H}$.

2. At line 13 on page 5, the authors describe “8Hmt transformed into an oxidized product O8Hmt and a further oxidized product OO8Hmt at the same time”. Do the two products (O8Hmt and OO8Hmt) are really generated at the same time? Figure 3 shows that OO8H is synthesized with the stepwise oxidation of 8H through O8H. The given description can also be interpreted as synthesizing OO8H from 8H directly. A more clear-cut expiation about this reaction is needed.

We thank Reviewer 1 for this important comment. We have monitored the reactions by TLC to gain more insights into the transformation process (only 1.0 equivalent of DDQ was added). The secondary oxidation product **OO8H_{mt}** with orange luminescence was clearly detected by TLC at the very beginning of the reaction (Figure for reviewer 4). Apparently, a competition exists between the oxidation of the starting helicene **8H_{mt}** and the oxidation of the newly formed preliminarily oxidized product **O8H_{mt}**. In contrast, **8H** solely transformed into the preliminary oxidation product **O8H** with the presence of 1.0 equivalent of DDQ. We have emphasized this point in the revised manuscript and SI.

Figure for reviewer 4. TLC monitoring for the oxidative cyclo-rearrangement of **8H_{mt}**. Dot 1 for the raw product and dot 2 for the starting helicene **8H_{mt}**.

3. Why did the helical chirality of **7H**, **OO8H**, **OO9H** have not been determined by X-ray crystallography? The performed collection of the full data set using CuK α radiation seems to be appropriate for absolute structure determination. Did the authors merge Friedel pairs in data processing? I requested the authors to explain the reason for it.

We have not merged the Friedel pairs for these crystals in the data processing. In fact, due to the low quality of the crystals, the diffractions were quite weak despite of multiple attempts, and the best possible data still showed relatively high R_1 and wR_2 values ($R_1 = 6.04\%$, 6.60% and 6.44% , and $wR_2 = 16.44\%$, 20.70% and 21.41% for **7H**, **OO8H** and **OO9H**, respectively). Besides, the lack of heavy atoms was also unfavorable for the determination of absolute structure. In consequence, the Flack parameters for **7H**, **OO8H** and **OO9H** were measured as $-2(11)$, $-10(16)$, $0(5)$, respectively, which showed large uncertainty. Therefore, although these crystals showed spontaneous resolution phenomenon in the crystallization, it is of high difficulty to determine the absolute structure.

4. In CheckCif outputs, level A or B alerts are found in **O9H**, **O8HCF3**, **OO9H**, **OO8H_{mt}**, and **O6HDPT**. The authors should address these alerts by writing vrf in CIFs.

We have added Author Response for the level A and B alerts of **O9H**, **O8H_{CF3}**, **OO9H** and **OO8H_{mt}** in the CIF files using vrf format. We have also solved the level B alert for **O6H_{DPT}**. The modified CIF files were re-uploaded and the corresponding description has been corrected in the revised SI.

5. Hall symbol in CIF of **8HF** and **OO8H_{mt}** is notified as unsuitable in CheckCIF. Modification or explanation (using vrf) is required.

We have solved the Hall group problem by changing the space group from $P2_1/n$ to $P2_1/c$. The modified CIF files were re-uploaded and the corresponding description has been corrected in the revised SI.

Reviewer 2:

This is a wonderful article that shows genuine novelty and remarkable significance to the field of curved polycyclic aromatics. The authors have found that oxidation of [n]helicenes ($n = 7$ to 9) with DDQ at presence of $\text{CF}_3\text{SO}_3\text{H}$ enables unprecedented skeletal rearrangement resulting pi-expanded [m]helicenes ($m = 5$ to 7). The pi-expanded [6]helicenes, such as O8H, can be further oxidized by DDQ/ $\text{CF}_3\text{SO}_3\text{H}$ giving cyclized products with formation of a seven-membered ring; while pi-expanded [7]helicenes, such as O9H, can be further oxidized by DDQ/ $\text{CH}_3\text{SO}_3\text{H}$ enabling another oxidative cyclo-rearrangement. A reasonable mechanism for this oxidation cyclo-rearrangement has been proposed on the basis of DFT calculations on the possible reaction pathways. Another very interesting finding is that the chirality of the helicene precursor is transferred to the resulting pi-expanded helicene with ee higher than 99%. Moreover, this is a very solid study having all the products unambiguously characterized. It is very impressive that 23 single crystal structure have been reported in this manuscript!

The Scholl reaction is a powerful method for synthesis of polycyclic aromatics enabling formation of multiple carbon-carbon bonds in a single step. However, synthetic organic chemists still lack general ability to accurately predict whether and how the Scholl reaction will occur on a new substrate. The oxidative cyclo-rearrangement under the Scholl reaction conditions discovered here has provided important new insights to the Scholl reaction.

Therefore, this manuscript is highly recommended for publication in Nature Communication. The following minor issues should be taken into consideration for revision.

We are very grateful for the reviewer's kind remarks and the following suggestions.

1. In Figure 1a, "Benzanullation" should be "Benzannulation".

We have corrected this issue in the revised manuscript.

2. OO8H is a chiral molecule and its single crystal structure contains only the M-enantiomer. Can the authors comment on the chirality transfer from O8H to OO8H in accompany with formation of a seven-membered ring?

The observation of a single enantiomer of **OO8H** in the crystal was actually a consequence of a rare spontaneous resolution process, i.e., self-sorting of two enantiomers of **OO8H** occurred during crystallization. In fact, the racemization barrier of **OO8H** is relatively low ($\Delta G_{\text{calc}}^\ddagger = 17.6 \text{ kcal}\cdot\text{mol}^{-1}$, 298 K, Supplementary Figure 112) and hence the molecule undergoes racemization at room temperature ($t_{1/2} \sim 10^0 \text{ s}$). Therefore, in solution, we could only obtain racemic **OO8H** at the end.

3. Regarding the proposed mechanism shown Figure 2d, the authors are strongly suggested to explain why the second 1,2-migration occurs via a homoallylic transition rather than a homobenzylic transition. This explanation should be included in the main text rather than in the Supporting Information (Figure S104)

As suggested by Reviewer 2, we have emphasized this point in the revised manuscript: "Notably, the migration might undergo through a homobenzylic pathway to afford less crowded outward product but this is probably disfavored by a relatively higher activation energy (Supplementary Figure 104)".

Reviewer 3:

The manuscript entitled ‘Oxidative Cyclo-Rearrangement of Helicenes into Chiral Nanographenes’, presents a novel skeletal reconstruction reaction in helicenes and affords straightforward access to an array of chiral polycyclic aromatic hydrocarbons.

This work could contribute to a renewed perspective of Organic Synthesis and have direct implications in overlapping fields, such as Materials Science. This kind of intramolecular rearrangements are usually poorly explored, and often remain anecdotic. Further study and understanding of these processes will provide more efficient synthetic tools. The authors provided solid evidence for their claims. Particularly, I appreciate that X-ray structure of several compounds are included.

The manuscript is clear and well written. In my opinion, it should be published in Nat. Commun. once the following minor points have been addressed or considered.

We are very grateful for the reviewer’s kind remarks and the following suggestions.

Introduction

(1) In my opinion Aryne Chemistry is among the most used methods for the synthesis of structurally complex polycyclic aromatic hydrocarbons (PAHs) and helicenes. Therefore, I would suggest the addition of an explicit mention to that approach in the introduction.

We agree with reviewer 3. Aryne chemistry is indeed very important for PAH synthesis (*Acc. Chem. Res.* **2019**, *52*, 2472). We have addressed this point in the revised manuscript.

(2) In the sentence “or deform into planar structures through oxidative cyclization” I would add “nearly planar”, given that reference 20 is entitled “On-surface synthesis of a nonplanar porous nanographene”.

We have corrected the expression in the revised manuscript.

Results and discussion

(3) At the beginning of this part, authors explain how they discovered this interesting reaction, although there is not a clear optimisation. I would like to know how temperature, time, and the use of other solvents, affect to the oxidative cyclo-rearrangement (It would be great the addition of a table with few entries in the supporting information).

We thank Reviewer 3 for this important comment. Regarding the influence of helicene precursor and acid, we have supplemented a detailed screening of reaction condition in the revised SI (also see the response to Reviewer 1). In addition, we have also investigated the effect of temperature, reaction time and solvent on the oxidative cyclo-rearrangement. Generally, (i) in the presence of $\text{CF}_3\text{SO}_3\text{H}$, the reaction underwent well at 0°C but required a longer reaction time (Entry 1 vs. Entry 2); (ii) in the presence of $\text{CH}_3\text{SO}_3\text{H}$, a larger amount of DDQ (5 equiv.) along with a lower temperature substantially enhanced the yield of **O8H** (Entry 3 vs. Entry 5), but the helicene precursor mostly decomposed at a higher reaction temperature or upon a long reaction time (Entries 4 and 6); (iii) the reaction also underwent well in CHCl_3 and CH_3NO_2 (Entries 7 and 8). These results have been added to the revised SI.

Table for review 2. Oxidative cyclo-rearrangement of **8H** under various conditions.

Entry	Acid	DDQ	Temperature	Time	Solvent	Yield ^a
1	CF ₃ SO ₃ H	1.2 equiv.	25 °C	30 min	CH ₂ Cl ₂	49%
2	CF ₃ SO ₃ H	1.2 equiv.	0 °C	12 h	CH ₂ Cl ₂	48%
3	CH ₃ SO ₃ H	1.2 equiv.	25 °C	30 min	CH ₂ Cl ₂	Trace of O8H with a string of unidentifiable products
4	CH ₃ SO ₃ H	5 equiv.	25 °C	30 min	CH ₂ Cl ₂	Mostly decomposed
5	CH ₃ SO ₃ H	5 equiv.	-20 °C	30 min	CH ₂ Cl ₂	41%
6	CH ₃ SO ₃ H	5 equiv.	-20 °C	12 h	CH ₂ Cl ₂	Trace of O8H , mostly decomposed
7	CF ₃ SO ₃ H	1.2 equiv.	25 °C	30 min	CHCl ₃	39%
8	CF ₃ SO ₃ H	1.2 equiv.	25 °C	30 min	CH ₃ NO ₂	38%

^aIsolated yield after purification by column chromatography on silica gel.

(4) “Mass spectroscopy” should be corrected to “Mass spectrometry”.

We have corrected this issue in the revised manuscript.

(5) In the second paragraph of page 4. The authors say “Besides, both OO7H and OO8H can be directly synthesized in one pot starting from 7H and 8H in the presence of more DDQ.” I would include the yield for each one pot transformation in the main text. These relevant results are currently available in the supporting information (46 and 40% respectively).

We have added the yield for the one-pot transformation in the revised manuscript.

(6) In the last paragraph of page 5. In the sentence “The oxidative cyclo-rearrangement reactions were ultimately associated with other reactions to fabricate more sophisticated nanographenes with distinctive geometries and tailored properties.” I would delete tailored.

We have removed “tailored” in the revised manuscript.

(7) In the same paragraph, the word “nanoribbon” it is used referring to a part of compound O6HDPT. From my point of view, it would be more appropriate the use of another term, such as structure.

We have changed the word “nanoribbon” to “structure” in the revised manuscript.

References

(8) Reference 31. The name of the last author should be corrected (Campaña, instead of Caompaña).

We have corrected this issue in the revised manuscript.

Figure 2b

(9) I would replace “Unidentical” Products with “Unidentified” Products in both cases.

As suggested by Reviewer 3, we have changed “Unidentical” to “Unidentified” in the revised manuscript.

Supporting Information

(10) The characterisation provided is enough to strongly support the work. However, they should report which technique was used for mass analysis in each case (MALDI, APCI, EI, etc.).

We thank Reviewer 3 for this careful comment. MALDI mode was used for the mass analysis, and we have denoted this point in the revised SI.

REVIEWERS' COMMENTS

Reviewer #1 provided remarks only to the editor, in which they recommend publication.

Reviewer #3 (Remarks to the Author):

The authors fully addressed my concerns. I am glad to see the optimisation table added in the SI. Having considered the manuscript in the current form and the reply to the other referees, I would like to recommend its publication.